# Experimental Study on the Preparation of Recycled Admixtures by Using Construction and Demolition Waste

**DOI:** 10.3390/ma12101678

**Published:** 2019-05-23

**Authors:** Shujun Li, Qiuyi Li, Xiaolong Zhao, Jianlin Luo, Song Gao, Gongbing Yue, Dunlei Su

**Affiliations:** 1School of Civil Engineering, Collaborative Innovation Center of Engineering Construction and Safety in Shandong Blue Economic Zone, Qingdao University of Technology, Qingdao 266033, China; qllishujun@163.com (S.L.); lawjanelim@qut.edu.cn or j.luo@westernsydney.edu.au (J.L.); gaosong727@126.com (S.G.); sudunlei@163.com (D.S.); 2School of Architecture Engineering, Qingdao Agricultural University, Qingdao 266109, China; yuegongbing@163.com; 3China Academy of Building Research, Beijing 100013, China; 4Center for Infrastructure Engineering, School of Computing, Engineering and Mathematics, Western Sydney University, Sydney, NSW 2751, Australia

**Keywords:** Construction and demolition waste, recycled powder, basic properties, compatibility of cement and superplasticizer, microstructure

## Abstract

The use of construction and demolition waste (CDW) to prepare recycled admixtures is of great significance for the complete resource reutilization of CDW. In this paper, different kinds of CDW were prepared into recycled powder (RP) with a specific particle size (0–45 µm or 0–75 µm). The fineness, water requirement ratio (WRR), fluidity, loss on ignition (LOI), strength activity index (SAI) and compatibility of cement and superplasticizer (CCS) were examined. The above test results were analyzed by advanced analysis tools, such as laser particle size analysis, XRD, XRF, DSC-TGA, SEM, and BET. The properties of different types of RPs varied greatly, which was closely related to the microstructure, particle morphology and chemical composition of the RP. The experimental results showed that all kinds of RPs after grinding had a high fineness and good particle size distribution, and the mineral composition was dominated by SiO_2_ with the content exceeding 50%. The WRR of various RPs was between 105% and 112%, and the SAI was between 68% and 78%, but the LOI varied greatly. Different types of RPs had a negative impact on the CCS, but the compatibility of cement and naphthalene-based superplasticizer was less affected. The content of recycled brick powder (RBP) in a hybrid recycled powder (HRP) was an important factor. When the content of RBP in HRP exceeded 50%, the HRP could meet the basic performance requirements of fly ash.

## 1. Introduction

A series of treatments for construction and demolition waste (CDW) allow it to be reused in a building, which not only helps conserve natural resources but also solves the ever-increasing crisis of CDW disposal. Recycling has social, economic, and environmental benefits and profound significance for the sustainable development of cities [1].

In many countries and regions, recycled aggregate (RA) has been thoroughly and comprehensively researched [2,3,4,5]. For example, China has developed extensive standards for RA [6,7]. However, a portion of the CWD is a powder that is produced during the production and processing of RA. If not utilized, it will cause serious air pollution and endanger public health [8]. As an important building material, clay bricks are widely used. Currently, due to the rapid aging and the service life of many old buildings [9], the research on the reutilization of waste clay bricks has become very urgent. However, many studies [10,11] have demonstrated that the properties of aggregates prepared from waste clay brick are not ideal, and thus, researchers have looked in other directions.

The main components of CDW are concrete and ceramic materials, such as clay bricks [12,13]. Therefore, according to the composition of CDW, the research on RP can be roughly divided into three categories. The first type is a hybrid recycled powder (HRP) produced during the process of producing RA: Its raw materials include waste concrete and clay bricks. The aggregate debris, hardened cement paste and clay bricks in the powder each occupy a certain proportion [14]. The second type is a recycled concrete powder (RCP) produced in the reshaping process of RA. Its main component is aggregate crumbs and a small portion of hardened cement paste [15]. The third type is a recycled brick powder (RBP), which is prepared by mechanically pulverizing and grinding clay bricks [16].

At present, RP is mainly studied as supplementary cementitious material, similar to fly ash (FA) and ore powder. HRP contains a certain amount of active materials, such as unhydrated cement particles and a high content of SiO_2_, so it exhibits favorable activity when HRP reaches a certain fineness [17]. When the Portland cement is replaced by 30% HRP in concrete, the compressive strength, bending strength, and tensile splitting strength can be satisfied, but the anti-cracking performance is not ideal [14]. In addition, the chloride permeability and shrinkage decrease with increasing HRP content in concrete [18]. RBP reaching 75 µm or finer can exhibit good activity. Thus, concrete can still maintain good strength when RBP replaces 20% of the cement [19,20,21]. Furthermore, RBP can improve the pore structure of cement-based materials, which is beneficial to the improvement of concrete durability [22]. It can also inhibit the alkali–silicon reaction that may occur in concrete [23]. RCP is a low-activity powder that is produced in the reshaping process of RA. The addition of RCP generally increases the water consumption of concrete. However, when the cement substitution rate is less than 10%, RCP will contribute to the impermeability and the early strength development of the concrete. In addition, anti-carbonization requirements can be met when the proportion of RCP in cement is within 30% [15].

By comparing the results of some studies [24,25,26], it is found that the properties of RP are closely related to their composition. Considering the diverse composition of CDW, the research on the performance of RP should be classified and comprehensive, which can help people better understand the properties of RP. In addition, it is very important to combine experiments with existing, relatively mature CDW recycling systems. Researchers need to pay attention to what kind of interaction will occur when different components of RP are used together. The application of superplasticizers (SPs) in the construction industry has been very widespread. While maintaining the workability and improving the strength and durability of concrete, SPs can reduce water consumption [27]. Therefore, the influence of RP on the compatibility of cement and SPs (CCS) is worth studying.

To make the test as comprehensive as possible, we prepared three kinds of RPs. The powder generated by the process of reshaping RA is defined as RCP, and the powder prepared with waste clay brick and mortar blocks is defined as RBP and recycled mortar powder (RMP), respectively. To perfect the performance study of RP as much as possible, we selected two particle size ranges (0~75 µm and 0~45 µm) to discuss the influences of different RP particle sizes on its performance. In addition, we combined different kinds of RP together to discuss the relationship between the properties and components of HRP.

In this paper, the basic properties of RPs with different types and fineness were studied in detail, including fineness, particle size distribution, chemical, and mineral composition, loss on ignition (LOI), water requirement ratio (WRR), fluidity and strength activity index (SAI). In addition, the influence of RP on the CCS was discussed. Advanced tools, including laser particle size analysis, XRD, XRF, DSC-TGA, SEM, and BET, were also used to analyze the experimental results to better understand the influence of the microstructure of RP on the properties and to provide certain a theoretical reference for the practical application of RP in the future.

## 2. Materials and Methods of Experiment

### 2.1. Preparation Procedures of RP

There were three types of RPs used in the experiment. The first type came from the process of reshaping RA. The raw materials used were Grade-II RA produced by Qingdao LvFan Recycled Building Materials Co. Ltd. (Qingdao, China). The basic performance indexes of RA are shown in Table 1. The preparation process is shown in Figure 1. The instruments used in the experiment were a jaw crusher (Y160L-6, Linyi, China), a sieve machine (ZBSX-92A, Cangzhou, China), an RA reshaping machine (Qingdao, China), and a ball mill (YXQM-2L, Changsha, China, rotating speed: 400 r/min). The second type was RBP, and its raw material was waste clay brick from a construction demolition site in Qingdao, China. The preparation process of RBP is described as follows. First, the waste clay brick was cleaned to remove mortar. Then, the jaw crusher was used to crush the waste clay brick to obtain particles with particle sizes < 5 cm. Finally, RBP meeting the test requirements was prepared by a ball mill. The third type was RMP, and its raw material was a mortar block (age: 16 months, 28-d compressive strength: 5.4 MPa) prepared in the laboratory. The mixing mass ratio is 1:8:0.5 (cement:sand:water). The preparation of the RMP process was the same as RBP.

### 2.2. RP Mixing Proportion Design

To investigate the influence of component changes on HRP properties, different mixing proportion designs of HRP were used in the experiment as shown in Table 2. In recent years, China has been carrying out urbanization construction, and many rural areas have produced a large amount of CDW from masonry structures. This CDW is mainly composed of clay bricks and mortar. In the experiment, the combination of RBP and RMP is to simulate RP prepared from masonry CDW. The combination of RBP and RCP is to simulate RP prepared from brick–concrete structure CDW.

### 2.3. LOI Experiment of RP

LOI is an important index for classifying the quality of FA. The method of determining LOI [28] of RP is described below. The sample was dried to a constant weight at 105 ± 5 °C. Then the sample weighed to approximately 1 g (accurate to 0.0001 g), was burned to a constant weight at 950 ± 25 °C. The LOI was determined by the ratio of the weight after burning to constant weight to the weight before burning.

### 2.4. Fluidity and WRR Experiment

The fluidity and WRR can complement each other to evaluate the quality of an admixture. The fluidity test process [29] is shown in Figure 2. The weight proportion of the mortar is cement:RP:sand:water = 0.7:0.3:3:0.5. The instruments used in the experiment had a cement fluidity electric jumping table (MLD-3 type, Cangzhou, China) and cement mortar mixer (JJ-5 type, Wuxi, China). In the WRR experiment [30], the mixing proportion of the mortar was similar to that used in the fluidity test. The WRR of RP was determined by the ratio of the fluidity of mortar mixed with RP and a reference mortar without RP substitution. The water used was tap water. The cement, acquired from SUNNSY Co. Ltd. (Jinan, China), was the ordinary Portland cement, and its basic performance index is listed in Table 3.

### 2.5. SAI Experiment of RP

To investigate the change in the SAI of various RPs, the mixing proportion design shown in Table 4 was used in the experiment. The particle size range of RBPI, RMPI, and RCPI were 0–45 µm and the particle size range of RBPII, RMPII, and RCPII were 0–75 µm in Table 4. The SAI experiment of RP referred to the SAI experiment of FA [30]. The SAI of RP was determined by the ratio of the 28-d compressive strength of the mortar mixed with RP that of to the reference mortar without RP substitution.

The mortar was poured into molds with sizes of 40 × 40 × 160 mm^3^ for casting forming and under the standard curing room (T = 20 + 2 °C, RH ≥ 95%) maintenance, and the compressive strength of the specimen was measured after 28 days.

### 2.6. Experiment of RP Impact on CCS

CCS is defined as the degree of change in the fluidity of cement paste over time due to the quality of the cement and SPs and the degree of change in SP consumption when the same fluidity is obtained. The saturation point of SPs at which the fluidity of the cement paste was not increased with an increase in the SP content in the cement paste is an important indicator in CCS. The SPs used in the experiment were naphthalene-based SPs (NS) and polycarboxylate-based SPs (PCS). The specific properties are shown in Table 5. The weight proportion of cement paste is water:cement:RP = 0.29:0.7:0.3. The specific steps of the experiment [31] are shown in Figure 3.

To facilitate a discussion of the fluidity loss of cement paste mixed with RP over time, the loss rate of fluidity over time (LRFT) is introduced here, and the formula is as follows (1):(1)$$FL=\frac{T_{n}-T_{in}}{T_{in}}\times 100\%$$

In the formula:*FL*: LRFT, expressed as a percentage (%);*T_in_*: initial fluidity (mm);*T_n_*: n hours fluidity (mm).

The result is expressed to a single decimal place.

## 3. Results and Discussion

### 3.1. Fineness and Particle Size Distribution Analysis

Figure 4 shows that the fineness of various RPs is are linearly correlated with the grinding time. Under laboratory conditions, RP with a fineness (45 µm sieve) of 0% can be obtained by ball milling. This indicates that a good fineness of RP can be obtained by mechanical grinding. In addition, the grinding energy consumption of different kinds of RPs is different, which provides guidance for selecting a grinding system. If raw materials are sorted when RP is prepared, the grinding system may be single. If it is not sorted, it is necessary to use a high-efficiency screen in closed-circuit grinding to reduce energy consumption.

The particle size distribution of an admixture can reflect its gradation. Figure 5 shows the results of the laser particle size analysis of various RPs (Rise-2006 type, Jinan, China). Figure 5 shows that each type of RP has a particle size distribution similar to Grade-II FA, indicating that various RPs have a good particle size distribution. The particle size distributions of various RPIs with a fineness of 0% (45 µm screen) show that the particle size distribution of various RPIs becomes wider, and the differential distribution peak shifts to the left, which demonstrates that the RP gradation is improved [32]. Therefore, mechanical grinding can improve the gradation of RP.

### 3.2. Chemical and Mineral Compositions

X-ray fluorescence spectrometry (XRF, XRF-1800 type, Shimadzu Corp., Kyoto, Japan) and X-ray powder diffraction (XRD, D8 Advance type, Bruker Corp., Karlsruhe, Germany) were used to determine the chemical and mineral compositions of various RPs. Table 6 shows that the SiO_2_ content of the various RPs is close to that of FA, and the contents of CaO, Al_2_O_3_ and Fe_2_O_3_ are between those of the cement and fly ash, indicating that RP has a good oxide distribution. Figure 6 shows what kinds of RPs contain SiO_2_ crystal peaks. It has been reported that fine SiO_2_ can improve the compactness and mechanical strength of concrete [33]. In addition, the crystal peak of albite that is mostly found in granite is found in the XRD pattern of RCP, so the SiO_2_ in RCP originated mostly from aggregate debris. RCP also contains CaCO_3_ (calcite), which can shorten the induction period of C_3_S and partially participate in the hydration process of C_3_S [34,35]. However, the crystal diffraction peaks of calcium silicate and calcium aluminate are not found in the XRD patterns of RCP and RMP, indicating that the cement particles in RCP and RMP have been basically hydrated and mainly exist in the form of a gel, as shown in the red area in Figure 7. In the XRD pattern of RBP, only the crystal peak of SiO_2_ is obvious, indicating that the main mineral composition of RBP is SiO_2_.

### 3.3. LOI

Figure 8 shows that the LOI of RBP is the smallest, approximately 2%, followed by RMP and RCP at approximately 6% and 12%, respectively. However, the LOI of various RPs of the same kind with different fineness differs very little, within approximately 1%, which is different from FA. In addition, the LOI of HRP linearly decreases as the content of RBP in HRP increases.

Figure 9 shows the DSC-TGA (SDT Q600 type, TA Instruments, USA) analysis results of various RPs. To clarify the test results, the deriv.weight curve was used to express the test results. Figure 10 shows that the deriv.weight curve of the RCP decreases steadily before 700 °C. Combined with the analysis results of the chemical and mineral components, it can be determined that the weight loss of RCP before 700 °C was caused by the thermal decomposition of gels in the RCP [15,36]. Between 700 °C and 800 °C, CaCO_3_ (calcite) in RCP is thermally decomposed to release CO_2_. This part of the weight loss accounts for approximately 31% of the total weight loss of RCP. Therefore, the LOI of RCP is mainly due to the thermal decomposition of gels and calcite. Calcite in RMP is thermally decomposed between 700 °C and 750 °C, accounting for approximately 11% of the total weight loss of RMP. Under 700 °C, it is similar to RCP, in which the gel is heated and dehydrated. Although the deriv.weight curve of RBP has some peaks, the overall weight loss is stable. In summary, RMP and RCP have a large LOI, which is mainly caused by the thermal decomposition of gels and calcite in the powder, while RBP is a powder with a smaller LOI.

### 3.4. Fluidity and WRR

Figure 10 displays the test results of the WRR of various types of RPs. Figure 10 shows that RBP has a minimum WRR of approximately 105%, followed by RMP and RCP at approximately 107% and 111%, respectively. However, the WRR of various RPs is higher than that of Grade-II FA (fineness of 26%).

In view of the fact that RP contains a certain amount of gels, the specific surface area of RP was determined by the nitrogen adsorption method instead of the air permeability method. The nitrogen adsorption method is more suitable to determine the internal and external through-hole area of porous materials than the air permeability method [37]. In the experiments, a specific surface area tester (ASAP 2000 type, Micromeritics Instruments Co. Ltd., Atlanta, USA) and electron microscopy (JSM-7500F type, JEOL Co. Ltd., Japan) were used to measure various RPs. Figure 11 shows that the specific surface areas of different types of RPs vary greatly, and this difference mainly comes from the number of gels in RP. It has been reported in research that gel has a large specific surface area between 100,000 and 700,000 kg/m^3^ [38]. The large difference in the specific surface area results in different WRRs for different types of RPs. In addition, as the fineness of RP decreases, the specific surface area of RP increases, which explains the increase in WRR of RPI. The microscopic morphology of various RP particles shows that the particle shape of RPII is irregular with polygonal corners and notches, and many fine particles are attached to the surfaces of large particles, as indicated by the red area in Figure 12. In Figure 12, the green area shows that the particle morphology of RPI is obviously improved, the number of corners is reduced, and the shape is gradually changed from an irregular shape to a spheroidal shape. Moreover, the number of fine particles in the RP increases, as shown in the yellow area of Figure 12, and the gradation of the RP is optimized which explains the phenomenon that the difference between the WRR of RPI and RPII is very small. However, the particle morphology of both RPI and RPII is different from that of FA particles. The particles of fly ash are spherical with smooth surfaces, which can reduce water consumption in concrete [39,40]. The RP particles are mostly irregular polygons, so the WRR of RP is higher than that of FA with same fineness.

Figure 13 presents a diagram of the effect of RP on the fluidity of the mortar. Figure 13 shows that the mortar mixed with RBP has the highest fluidity, followed by the mortars mixed with RMP and RCP. The fluidity of the mortar mixed with HRP increases as the content of RBP in HRP increases. When the RBP content in HRP exceeds 50%, the fluidity is between Grade-II and III FA. The same kind of RP has little effect on the fluidity of mortar. The results of the RP fluidity test are consistent with the WRR test of RP.

### 3.5. SAI of RP

The SAI is an important index to measure the activity and quality of an admixture. To eliminate the activation energy brought by pulverization, all RPs after pulverization were set aside for a period of time, and then the SAI test was conducted. The SAI and compressive strength test results of various RPs are shown in Figure 14a,b. Figure 14a shows that the SAI of RMP and RCP is between 68% and 72%. The SAI of RBP is better than that of RMP and RCP, and the SAI values of RBPII and RBPI are 74% and 78%, respectively, which is consistent with the results of a previous study [41]. Figure 15 shows that, compared with the mortar block mixed with RCP and RMP, the ettringite in the mortar block mixed with RBP is denser, and the pores are smaller, so the compressive strength of the mortar blocks mixed with RBP is higher [22,42]. Meanwhile, Figure 14a also shows that with the increasing RP fineness, the SAI values of different types of RPs increase. The reason is that as the fineness of RP increases, the number of tiny SiO_2_ crystals in the RP gradually increases, as shown in the yellow area in Figure 12. These fine SiO_2_ crystals can not only improve the microstructure and density of the mortar but also promote pozzolanic reaction [43,44].

The fitting results of the SAI of HRP show that with the increasing RBP content, the SAI of HRP is continuously increasing. When the RBP content exceeds 50% in HRP, the SAI of HRP is higher than 70%, which can meet the minimum SAI requirements for FA in China.

### 3.6. The Influence of RP on CCS

The CCS experiment mainly measured the saturation point of SPs (NS and PCS) and the fluidity loss of cement paste mixed with RP over time, as shown in Figure 16 and Figure 17, respectively. To better reflect the influence of RP on CCS, the FA group was not added here, but the pure cement group was added. Figure 16a shows that the saturation point of NS (SPNS) in pure cement paste is approximately 0.8%, and the SPNS of the cement slurry mixed with RBP is approximately 1%. The SPNS values of the cement paste mixed with RMP and RCP are much higher than that of pure cement paste at approximately 1.4% and 1.8%, respectively. Figure 16b shows that the saturation point of PCS (SPPCS) in pure cement paste is approximately 0.09%, followed by the SPPCS values of cement paste mixed with RBP at approximately 0.12%. The SPPCS of cement paste mixed with RMP is approximately 0.14%, and the maximum SPPCS of cement paste mixed with RCP is approximately 0.18%. Analysis of the above results shows that the RP in cement paste raises the saturation point of NS and PCS, but the effect of RMP and RCP is even more pronounced. There are two reasons for this: First, the particle morphology of RP is poor. When replacing cement as an admixture, this particle morphology will definitely reduce the fluidity of cement paste at the same water-to-binder ratio. Therefore, cement paste mixed with RP will consume more SPs to achieve the same fluidity as pure cement. Second, RP has a high specific surface area. Studies [45] have shown that admixtures with a high specific surface area can absorb more surface water and SPs, which can reduce the amount of free water and the effective SPs in the liquid phase, reduce the fluidity of the paste. Therefore, the fluidity of cement paste mixed with RCP > the fluidity of cement paste mixed with RMP > the fluidity of cement paste mixed with RBP > the fluidity of cement paste.

The LRFT is an important index to describe the CCS. The interaction between SPs and binder particles (such as silica fume or fly ash) may be affected, which means that the compatibility between mineral addition needs to be studied [46]. Figure 17a is the columnar accumulation diagram of the LRFT of the fluidity of cement paste mixed with RP. The experimental selection point is the saturation point of NS and PCS in cement paste. A three-dimensional diagram is selected here for expression, as shown in Figure 17b, so that the change in the test results is clearer. Figure 17a shows that various RPs in cement paste have a negative influence on 1 and 2 h LRFT. The LRFT (PSC) is affected more than the LRFT (NS) as shown in Figure 17b. In addition, RBP in cement paste has less influence on LRFT than RMP and RCP. There are two reasons for this: First, the work of SPs in cement paste is a continuous process. RCP and RMP have a large specific surface area and thus continuously adsorb SPs and water in the liquid phase over time, resulting in a decrease in the available SPs and water content in the system. Thus, the fluidity of the cement paste mixed with RP is continuously reduced. Second, compared with NS, PCS has the advantage of smaller doses amount, reducing water consumption, maintaining fluidity and decreasing concrete shrinkage of concrete [47], but PCS is very sensitive to changes in the cementing material [48]. Due to the high efficiency and sensitivity of PCS, the PCS content in the liquid phase is reduced faster than NS under the same adsorption conditions in cement paste, resulting in the LRFT being more significant.

## 4. Conclusions

Based on the chemical composition and microstructure of RP, its fineness, particle size distribution, LOI, WRR, SAI, and the effect of CCS were studied, and the following results and conclusions were obtained.

The particle size distribution of various RPs is similar to Grade-II FA. Through ball mill grinding, RP can obtain a good fineness, with 100% passing through a 45 µm sieve. Ball mill grinding can also improve the particle size distribution of RP, making the particle size distribution of various RPs better than that of Grade-II FA.Various RPs have a good oxide distribution, in which the SiO_2_ content is slightly higher than that in Grade-II FA, and the contents of CaO, Al_2_O_3,_ and Fe_2_O_3_ are between those of cement and Grade-II FA. However, there were significant differences in the LOI of various RPs. The LOI of RCP and RMP was large, and the LOI mainly came from the thermal decomposition of the gel and calcite in the powder, while the LOI of RBP was very small.The special particle morphology and large specific surface area of RP result in a higher WRR than that of Grade-II FA and a lower fluidity than that of Grade-II FA. The WRR is between 105% and 112%, and RBP is more favorable than the other two types of RPs to improvement of the mortar’s fluidity.RP contains a certain amount of fine SiO_2_ crystals and cementitious materials which impacts RP activity. The SAI of RCP and RMP is between 68 and 72% and the SAI of RBP can reach 78%. When the content of RBP in HRP exceeds 50%, the SAI can reach 70% or more.The large specific surface area of RP leads to a negative impact on the CCS, which is reflected in an increase in the saturation point of SPs and the 1-h and 2-h LRFT of the cement paste mixed with RP. Compared with RMP and RCP, RBP has a smaller surface area, so RBP has a small impact on CCS. In addition, RP has a much greater impact on the compatibility of cement with PCS than cement with NS.

The results presented in this paper show that there is great potential for the utilization of RP in mortar and concrete as a partial replacement for cement, in which the content of RBP in RP is an important factor.

Future studies will focus on the impact of RP on cement hydration and the adsorption behavior of RP on SPs, providing a theoretical reference for the practical application of RP.

## Figures and Tables

**Figure 1 materials-12-01678-f001:**
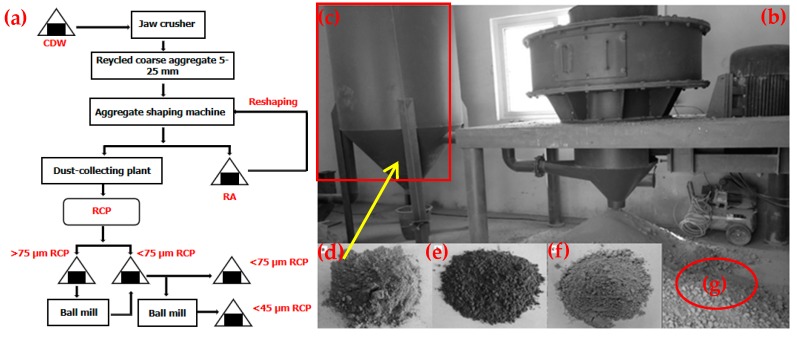
(**a**) Preparation process of recycled concrete powder (RCP), (**b**) reshaping machine, (**c**) dust-collecting device, (**d**) RCP, (**e**) recycled brick powder (RBP), (**f**) recycled mortar powder (RMP), and (**g**) reinforced RA.

**Figure 2 materials-12-01678-f002:**
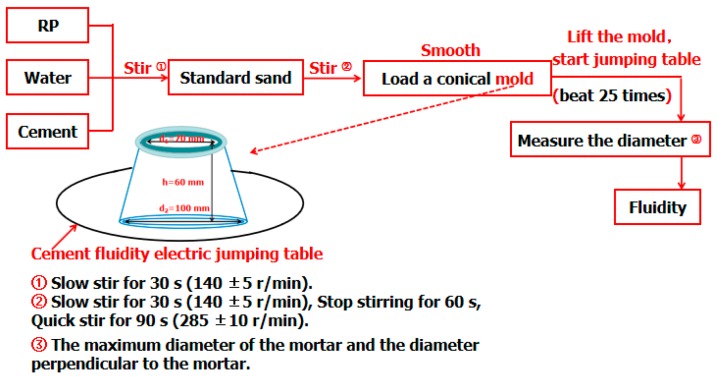
Fluidity and water requirement ratio (WRR) test process of RP.

**Figure 3 materials-12-01678-f003:**
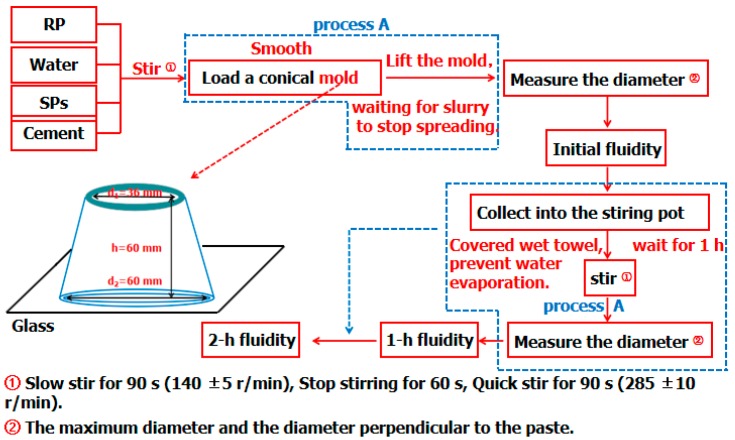
Experimental process of the RP impact on the cement and superplasticizer (CCS).

**Figure 4 materials-12-01678-f004:**
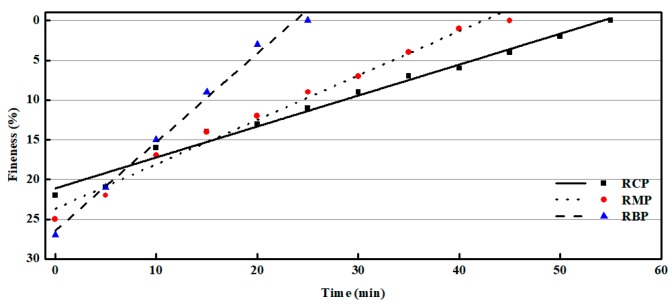
The influence of grinding time on the RP fineness.

**Figure 5 materials-12-01678-f005:**
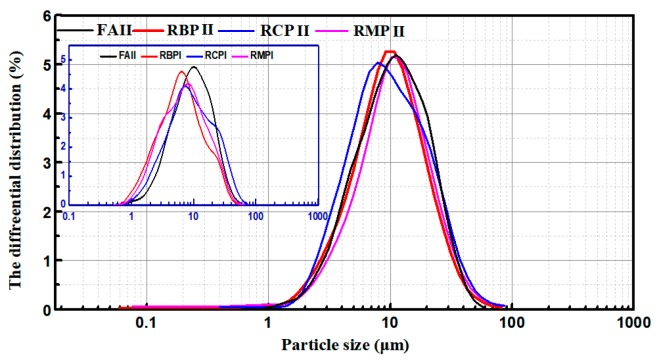
Particle size distribution of RP.

**Figure 6 materials-12-01678-f006:**
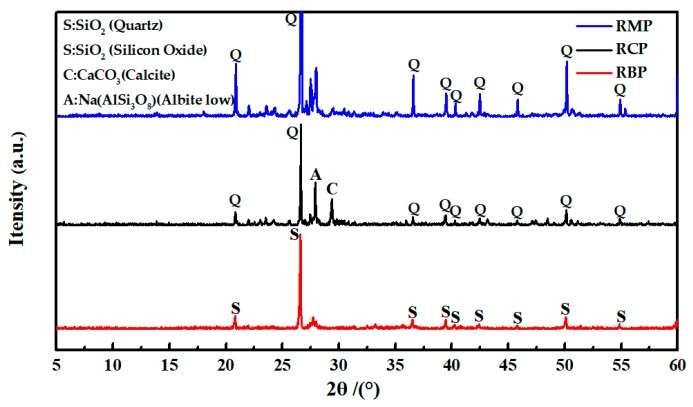
The XRD patterns of various RPs.

**Figure 7 materials-12-01678-f007:**
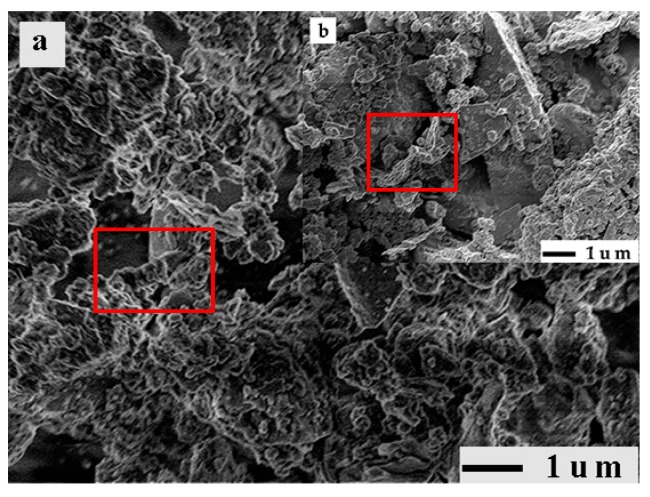
The fibrous gels in RPs: (**a**) RCP, (**b**) RMP.

**Figure 8 materials-12-01678-f008:**
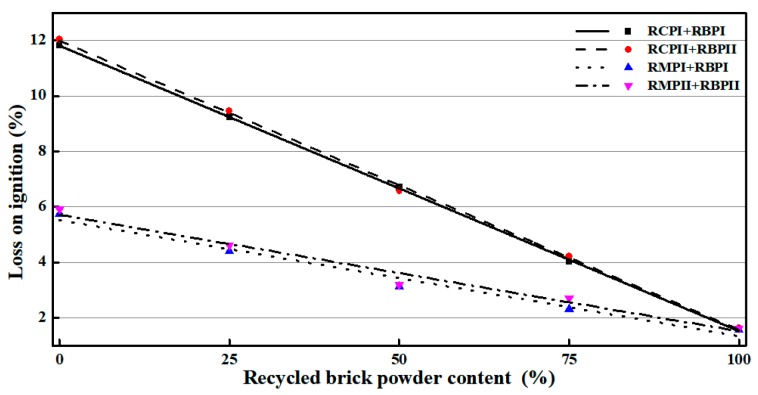
The LOI experiment results of various RPs.

**Figure 9 materials-12-01678-f009:**
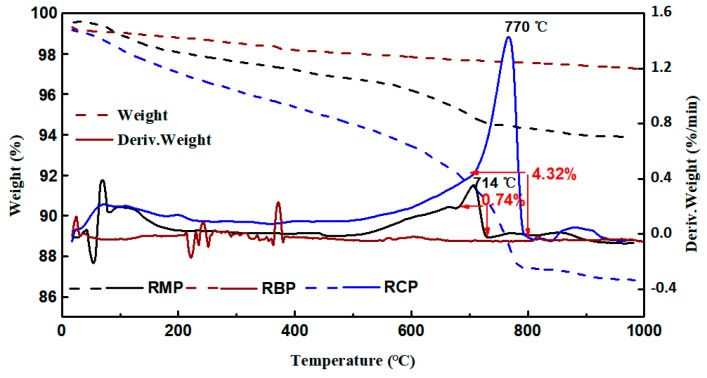
The TG-DTA test results of various RPs.

**Figure 10 materials-12-01678-f010:**
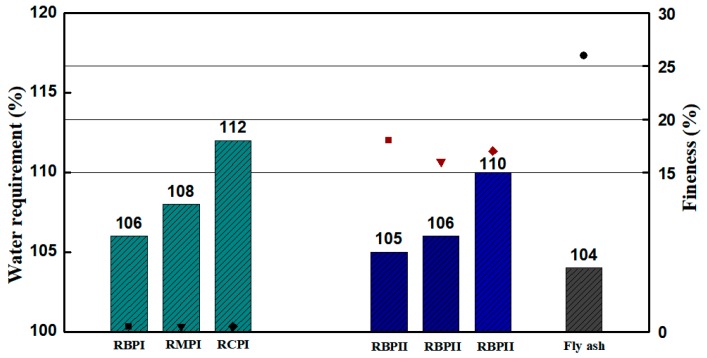
The WRR and fineness of RP.

**Figure 11 materials-12-01678-f011:**
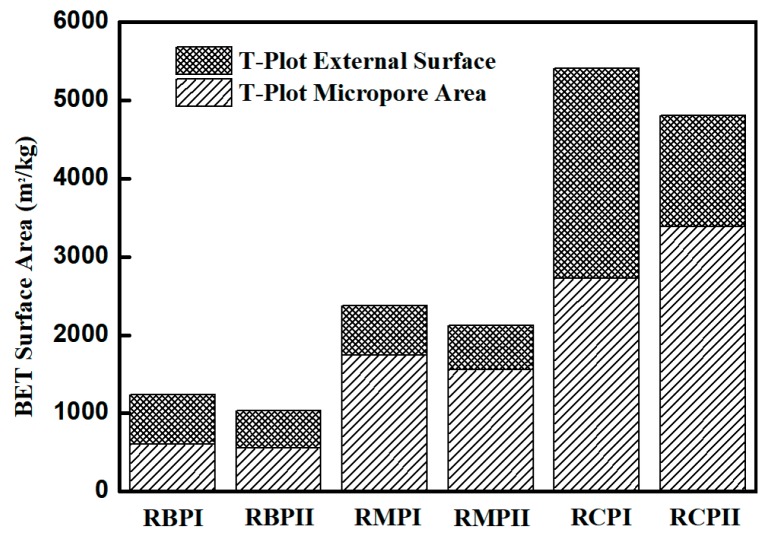
Surface area of various RPs.

**Figure 12 materials-12-01678-f012:**
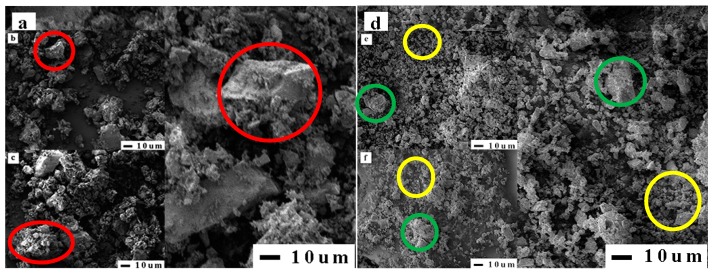
Microstructure of various RP particles: (**a**) RCPII, (**b**) RBPII, (**c**) RMPII, (**d**) RCPI, (**e**) RBPI, and (**f**) RMPI.

**Figure 13 materials-12-01678-f013:**
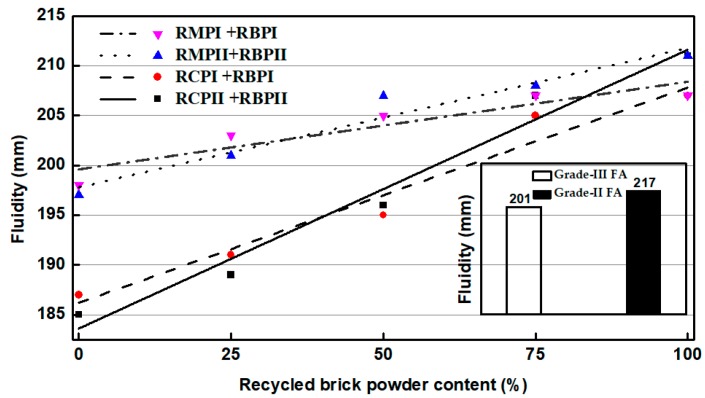
Results of the fluidity of mortar mixed with various RPs.

**Figure 14 materials-12-01678-f014:**
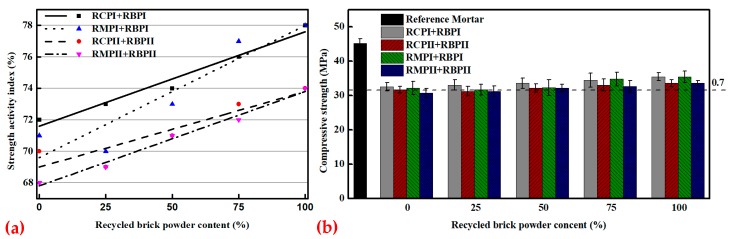
The results of the SAI experiment: (**a**) SAI, (**b**) compressive strength of mortar mixed with RP.

**Figure 15 materials-12-01678-f015:**
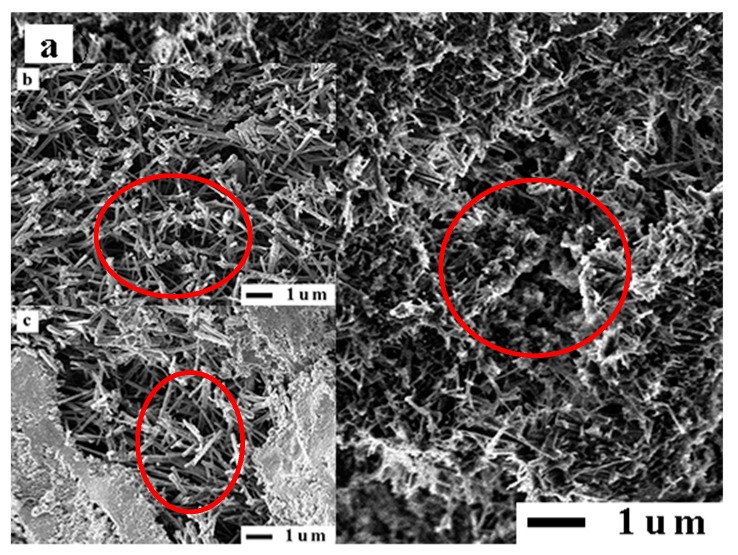
SEM morphology of pore structures of mortar mixed with RP: (**a**) RBP, (**b**) RMP, (**c**) RCP.

**Figure 16 materials-12-01678-f016:**
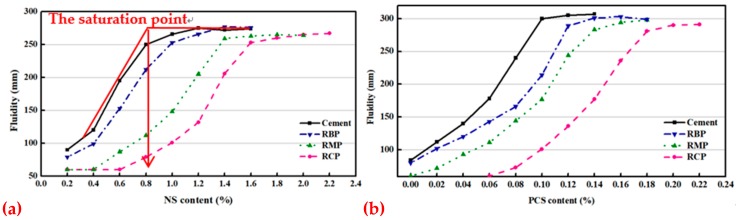
The saturation point of cement paste mixed with RP: (**a**) NS, (**b**) PCS.

**Figure 17 materials-12-01678-f017:**
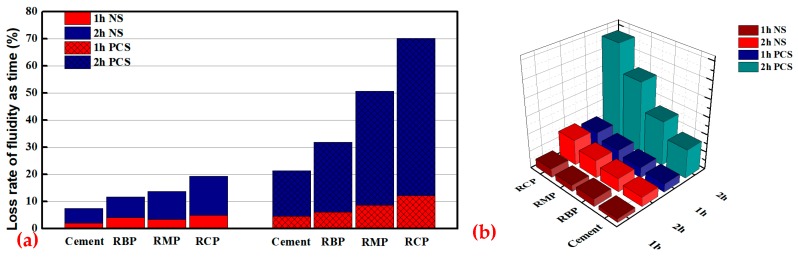
The LRFT of cement paste mixed with RP: (**a**) The columnar accumulation diagram of the LRFT, (**b**) The three-dimensional diagram of the LRFT.

**Table 1 materials-12-01678-t001:** The basic properties of recycled aggregate (RA).

Group	Water Content (%)	Water Absorption (%)	Apparent Density (kg/m^3^)	Bulk Density (kg/m^3^)	Crushing Index (%)
**RA**	1.21	7.6	2240	1300	26

**Table 2 materials-12-01678-t002:** Hybrid recycled powder (HRP) mixing proportion design **(%)**.

**HRP**	**RBP**	0	25	50	75	100
	**RMP/RCP**	100	75	50	25	0

**Table 3 materials-12-01678-t003:** Basic performance index of cement.

Cement Variety	Compressive Strength (MPa)		Flexural Strength (MPa)		Stability
	3 d	28 d	3 d	28 d	
**Portland cement**	19.1	46.6	4.7	7.6	**Qualified**

**Table 4 materials-12-01678-t004:** Mixing proportions of mortar (kg·m^−3^).

Cement	Types		Dosage		Substitution Rate (%)		Standard Sand	w/b
450	-	-	-	-	-	-	1350	0.5
315	RBPI	-	135	-	30	-		
	RBPI	RMPI	101.25	33.75	22.5	7.5		
	RBPI	RMPI	67.5	67.5	15	15		
	RBPI	RMPI	33.75	101.25	7.5	22.5		
	-	RMPI	-	135	-	30		
	RBPII	-	135	-	30	-		
	RBPII	RMPII	101.25	33.75	22.5	7.5		
	RBPII	RMPII	67.5	67.5	15	15		
	RBPII	RMPII	33.75	101.25	7.5	22.5		
	-	RMPII	-	135	-	30		
	RBPI	-	135	-	30	-		
	RBPI	RCPI	101.25	33.75	22.5	7.5		
	RBPI	RCPI	67.5	67.5	15	15		
	RBPI	RCPI	33.75	101.25	7.5	22.5		
	-	RCPI	-	135	-	30		
	RBPII	-	135	-	30	-		
	RBPII	RCPII	101.25	33.75	22.5	7.5		
	RBPII	RCPII	67.5	67.5	15	15		
	RBPII	RCPII	33.75	101.25	7.5	22.5		
	-	RCPII	-	135	-	30		

**Table 5 materials-12-01678-t005:** Physical and chemical properties of the superplasticizers (SPs).

SPs	Chemical	Appearance/Color	pH (20 °C)	Density (g/cm^3^)	Solid Content (%)
**NS**	**Sodium Naphthalene Sulphonate**	Brown powder	7.0 ± 1	450 ± 10	95 ± 1
**PCS**	**Modified Polycarboxylate**	White powder	8.0 ± 1	510 ± 10	98 ± 1

**Table 6 materials-12-01678-t006:** Chemical composition of fly ash determined by XRF **(w/%)**.

Constituent	SiO_2_	CaO	Al_2_O_3_	Fe_2_O_3_	MgO	K_2_O	Na_2_O	SO_3_
**Cement**	21.14	52.25	5.01	3.87	1.23	0.64	0.39	2.71
**Fly ash**	52.04	4.33	31.85	7.26	0.41	0.52	0.35	--
**RCP**	50.93	18.18	13.55	6.37	2.73	3.50	2.45	0.92
**RCP**	56.64	17.41	10.54	5.59	1.24	3.53	3.02	0.92
**RBP**	60.56	10.70	17.16	3.46	1.78	2.50	1.26	0.34

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
