# Peer review of "Experimental Study on the Preparation of Recycled Admixtures by Using Construction and Demolition Waste"

_materials, 2019, doi:10.3390/ma12101678_

Reviewer 1 Report

I must commend the authors of this work. The article is well written. There are only a few minor corrections, which I have identified in the annotated pdf attached to this report.                                                                                            

Author Response

First of all, the author hope that the respected review expert or editor can forgive the author for some language errors in his description.

Point 1 Please, revise the marked part of the fourth paragraph of the introduction.

Response 1: Delete “ cement ”. Delete “ RP ” and change it to “ RCP ”. Please, see the marked part in green in the article.

Point 2 : Please, revise the marked part of the first paragraph of the 2.2.

Response 2: Delete “ RBP and RCP ” and change it to “ RBP and RMP ” . Please, see the marked part in green in the article.

Point 3 : Please, revise the marked part of the first paragraph of the 2.4.

Response 3: Delete “ The mixing proportion of the mortar is 0.7:0.3:3:0.5 (cement:RP:sand:water). ” and change it to “ The weight proportion of the mortar is cement:RP:sand:water = 0.7:0.3:3:0.5. ”.

Please, See the marked part in green in the article.The modification is made by reference to some papers [1,2] of the same type. Here, the author thinks that this expression method can be more intuitive, clear description of the mixing proportion of mortar mixed with the recycled powder. The mxing proprotion of mortar is also defined more accuratelly.

If the review expert still have different Suggestions, the author hope the review to give an example. The author will revise it immediately.

1. Liu, Q.; Tong, T.; Liu, S.H.; Yang, D.Z.; Yu, Q. Investigation of using hybrid recycled powder from demolished concrete solids and clay bricks as a pozzolanic supplement for cement. Constr. Build. Mater. 2014, 73, 754–763.

“ The mix proportion for mortar samples is cement:recycled powder:sand:water = 0.7:0.3:3:0.5  ”

2. Pereira-de-Oliveira, L.A.; Castro-Gomes, J.P.; Santos, P.M. The potential pozzolanic activity of glass and red-clay ceramic waste as cement mortars components. Constr. Build. Mater. 2012, 31, 197-203.

" The mortar mixture were produced with the weight proportions of 1:3:0.5 (binder:sand:water)."

Delete “ The water was tap water, and basic the performance index of the cement (SUNNSY Co. Ltd, Jinan, China) is listed in Table 3. ” and change it to “ The water used was tap water. The cement , acquired from SUNNSY Co. Ltd. (Jinan, China), was the ordinary Portland cement, and it’s basic performance index is listed in Table 3. ” .

 Please, see the marked part in green part. If there is still anything inappropriate in this sentence, please point it out.

Point 4 : Please, revise the marked part of the second paragraph of the 2.5.

Response 4: Delete “ with ”. Please, see the marked part in green in the article.

Point 5 : Please, revise the marked part of the first paragraph of the 2.6.

Response 5: Delete “ Table 4 ” and change it to “ Table 5 ”. Delete “The mixing proportion of cement paste was 0.29:0.7:0.3 (water:cement:RP).” and change it to “ The weight proportion of cement paste is water:cement:RP = 0.29:0.7:0.3. ”. The mxing proprotion of cement paste is also defined more accuratelly.

Please, see the marked part in green in the article.

Point 6 : Please, revise the marked part of the second paragraph of the 3.4.

Response 6: Confirm that “ Figure 11 ” is correct. Delete “ 11 ” and change it to “ 12 ”. Please, see the marked part in green in the article.

Point 7 : Please, revise the marked part of the second paragraph of article 5 of the conclusion.

Response 7: Delete “ the SP of SPs ” and change it to “ the saturation point of SPs ”. Please, see the marked part in green in the article.

In addition, the author also re-examined the article. Please, see the marked part in purple in the article.

1. Revise the order of full-text reference.

2. The position of the mark in figure 1 has been adjusted.

3. Revise the font format in Table 1.

4. Revise the incorrect mark in Figure 13.

5. Revise the incorrect mark and subtitle of figure 17. Delete “ SP ” and change it to “The saturation point ”. Delete “ The SP ” and change it to “ The saturation point ”. Adjusted the position of the mark in Figure 17.

6. Revise formatting errors in some titles.

7. Revise formatting errors in references.

8. Confirm and revise the information of the authors.

If there are any other errors in this article that the author have not checked out, please let the author know. The author will revise it immediately.

Reviewer 2 Report

Please see the file attached

Author Response

First of all, the author hope that the respected review expert or editor can forgive the author for some language errors in his description.

Point 1 : Please, specify that FA means fly ash in Abstract.

Response 1: Delete “ FA ” and change it to “ fly ash ”. Please, see the marked part in yellow in the article.

Point 2 : Please, describe better the type of RP: RBPI/ II and RCPI/ II. In the text the difference is not highlighted.

Response 2: Added a description that “ The particle size range of RBPI, RMPI and RCPI were 0-45 µm and the particle size range of RBPII, RMPII and RCPII were 0-75 µm in Table 4. ” Please, see the marked part in yellow in the article.

Point 3 : Please, correct the section title numeration.

Response 3: Delete “ 2. Results and discussion ” and change it to “ 3. Results and discussion ”. Delete “ 3. Conclusion ” and change it to “ 4. Conclusion” Please See the marked part in yellow in the article.

Point 4 : Please, describe what the red area represents in Figure 6. There are more SiO2 peaks.

Response 4: To make the article more coherent, the red mark has delete.

Point 5 : Sub-section 3.5. Please, check the legend in Figure 14 (a). There seem to be two curves for RCPI + RBPI and RMPI + RBPI. 

Response 5: Delete the third and fourth legends and change it to “ RCPII + RBPII and RMPII + RBPII ”. Please, see the marked part in yellow in the article.

Point 6 : the introduction must be improved adding more references that could contribute to wider the state-of-the-art in literature in the field: 

Batayneh M, Marie I, Asi I. “Use of selected waste materials in concrete mixes”. Waste Manage 2007;27(12):1870–6.

D. Foti D, S. Vacca: “Comportamiento mecánico de columnas de hormigón armado reforzadas con mortero reoplástico./ Mechanical behavior of concrete columns reinforced with rheoplastic mortar”. Materiales De Construcción, Vol. 63, n. 310, pp. 267-282, abril-junio 2013, ISSN: 0465-2746, doi: 10.3989/mc.2012.03512.

D. Foti: “Innovative techniques for concrete reinforcement with polymers”, Construction and Building Mat. 2016, Vol. 112, pp. 202-209.doi:10.1016/j.conbuildmat.2016.02.111.

Response 6: The introduction has been revised. Please, see the marked part in yellow in the article. All articals have been quoted in the right places.

In addition, the author also re-examined the article. Please, see the marked part in purple in the article.

1. Revise the order of full-text reference.

2. The position of the mark in figure 1 has been adjusted.

3. Revise the font format in Table 1.

4. Revise the incorrect legends in Figure 13.

5. Revise the incorrect mark and subtitle of figure 17. Delete “ SP ” and change it to “The saturation point ”. Delete “ The SP ” and change it to “ The saturation point ”. Adjusted the position of the mark in Figure 17.

6. Revise formatting errors in some titles.

7. Revise formatting errors in references.

8. Confirm and revise the information of the authors.

If there are any other errors in this article that the author have not checked out, please let the author know. The author will revise it immediately.
